# Mechanisms Linking Social Media Use and Sleep in Emerging Adults in the United States

**DOI:** 10.3390/bs14090794

**Published:** 2024-09-09

**Authors:** Joshua Ethan Kinsella, Brian N. Chin

**Affiliations:** Department of Psychology, Trinity College, Hartford, CT 06106, USA; joshua.kinsella@trincoll.edu

**Keywords:** social media use, emotional investment, insomnia, sleep quality, social comparison, cognitive arousal

## Abstract

Social media use is associated with poor sleep outcomes. We aimed to extend previous research by examining how measures of social media use would affect two sleep characteristics: sleep quality and insomnia symptoms. In addition, we tested a serial mediation model linking social media use to sleep through increases in negative social comparison and pre-sleep cognitive arousal. Participants were 830 emerging adults (ages 18–30) who were recruited for an online survey study in February 2024. The sample was 63.1% female, with an average age of 24. We examined three measures of social media use: duration (minutes of daily use), frequency (number of weekly visits to social media platforms), and emotional investment (attachment to and integration of social media into daily life). Consistent with our hypothesis, greater emotional investment in social media and more frequent social media use were associated with poorer sleep quality and greater insomnia severity. We also found evidence supporting our hypothesized serial mediation model: emotional investment in social media and more frequent social media use were associated with increased negative social comparison, which subsequently increased pre-sleep cognitive arousal, which then led to poorer sleep outcomes. Our findings suggest that negative social comparison and pre-sleep cognitive arousal are important mechanisms linking social media use to poor sleep outcomes. Future studies should aim to test this serial mediation model using longitudinal data and experimental methods.

## 1. Introduction

Insufficient sleep remains a highly prevalent threat to young adults’ health, well-being, and success in the United States and worldwide [1]. Some studies have estimated that nearly two-thirds of young adults in the United States routinely get insufficient sleep [2] and report excessive daytime sleepiness [3].

Social media use is one common cause of poor sleep for young adults [4]. One study reports that 84% of young adults use some social media platform [5], and an ever-growing body of literature implicates social media use as an especially potent driver of poor sleep outcomes [6]. For example, studies have demonstrated that using social media close to bedtime was associated with greater sleep onset latency [4,7,8,9]. Nighttime social media use (i.e., the half-hour before bedtime) [10] and overall social media use [7,8] have also been linked with greater sleep disturbances. Moreover, nighttime social media use has been shown to contribute to later bedtimes, longer sleep onset latency, and diminished sleep duration [9].

### 1.1. Emotional Investment in Social Media

As a growing number of studies have explored the link between social media and sleep, there has been an increased focus on identifying the optimal way to assess social media use. Typical assessments of social media use in this literature have included duration (i.e., estimated average daily minutes of social media use) and frequency (i.e., estimated number of visits to social media per week) [7]. However, these simplified measures may be insufficient for capturing the more complex psychological impacts of social media use in context [9].

Woods and Scott [11] proposed that measuring emotional investment in social media may better capture the dimension of social media use that most strongly disturbs sleep. Emotional investment in social media may reflect the extent to which individuals feel connected to it and believe that using it is important to their daily lives. Indeed, Woods and Scott [11] found that emotional investment in social media was more strongly associated with poor sleep than overall social media use measures. Emotional investment in social media has also been linked to other negative mental health outcomes, including increased anxiety and depression [12] and lower self-esteem [11]. Thus, it is possible that emotional investment in social media may represent the key dimension of social media use associated with poor sleep.

### 1.2. Mechanisms Linking Social Media and Sleep

We considered negative social comparison and pre-sleep cognitive arousal as two possible mechanisms linking social media use and poor sleep.

According to social comparison theory, humans are driven by a need to evaluate their skills and abilities by comparing themselves to others [13]. The rise of social media has provided a new avenue for engaging in social comparison with others. Indeed, there is robust evidence for a bidirectional relationship between social media use and social comparison behaviors. Some studies have shown that social comparison orientation is positively associated with social media use [14,15,16], whereas others have shown that greater social media use [17] and greater passive consumption of social media [18] increase the tendency to engage in social comparison. In turn, engaging in negative social comparison has been linked with poorer well-being [19] and increased risk of loneliness [20]. Similarly, individuals higher with social comparison orientations report poorer self-perceptions, self-esteem, affect balance [16], and decreased positive affect [21]. In theory, each of these aforementioned correlates and the consequences of negative social comparison could be linked with heightened pre-sleep cognitive arousal.

Cognitive arousal caused by social media use may lead to poor sleep outcomes. Cognitive pre-sleep arousal has already been shown to mediate the relationships between binge-viewing television and poor sleep quality, insomnia (i.e., difficulty falling or staying asleep), and fatigue (i.e., daytime sleepiness) [22]. Numerous studies have identified an association between social media usage and sleep outcomes. Cognitive pre-sleep arousal mediated the relationship between significant social media use and daytime fatigue in people using TikTok [23] in an automatic fashion [24]. Additionally, a study by Harbard et al. [25] found that the relationship between online social media use and sleep disturbances was mediated by cognitive arousal before sleep. While it has been noted that social media may promote emotional, cognitive, and physical arousal [7], this arousal on a cognitive level may be caused by other underlying concerns such as fear of missing out (i.e., apprehension about missing out on rewarding experiences or enjoyable events) [9]. Other significant causes of pre-sleep cognitive arousal include worry and rumination [26], as well as social media algorithms and automatic scrolling practices that increase the likelihood of viewing more arousing media [24]. As we know, pre-sleep cognitive arousal can lead to poor sleep outcomes; it is important to further explore its causes and effects.

Measures of pre-sleep cognitive arousal assess how intensely an individual experiences psychological and somatic symptoms, such as racing thoughts, shortness of breath, racing heart, and perspiration while attempting to fall asleep. Levenson et al. [7] suggested that increased arousal may represent one mechanism linking social media use with disrupted sleep. Consistent with this possibility, numerous recent studies have identified cognitive pre-sleep arousal as a mediator of the associations of social media use with poor sleep and circadian outcomes [24,25,27]. Social media use, particularly at nighttime, is thought to increase pre-sleep cognitive arousal due to underlying psychological processes that include negative social comparison, fear of missing out [9], and increases in worry and rumination [26].

### 1.3. Aims and Hypotheses

This study tested the association between social media use and poor sleep in a sample of emerging adults (ages 18–30), and it investigated negative social comparisons and pre-sleep arousal as mediators of this association. We focused on emerging adults because of prior evidence for high rates of social media use and insufficient sleep in this population. First, we hypothesized that greater emotional investment in social media would predict poorer sleep quality and greater insomnia severity (Hypothesis 1). We predicted that effect this would occur above and beyond the effects of other measures of social media use, including the volume and frequency of social media use. Second, we hypothesized that negative social comparisons online and pre-sleep arousal would mediate the association between social media use and poor sleep. Specifically, we tested the mediation model shown in Figure 1, linking social media use to poor sleep, via simple mediation effects of negative comparisons on social media and pre-sleep arousal (Hypothesis 2a) and a serial mediating effect with negative comparisons on social media as the first mediator and pre-sleep arousal as the second mediator (Hypothesis 2b).

## 2. Materials and Methods

### 2.1. Participants and Procedures

Participants were recruited for a cross-sectional observational study of social media, sleep, and mental health in young adults. We determined our target sample size by conducting an a priori power analysis using G*Power version 3.1.9.7 [28] to determine the sample size required to test our first aim. We found that the required sample size to achieve 80% power for detecting a small effect in a bivariate correlation (*r* = 0.10) with two tails and an error probability of α = 0.05 was N = 800; this estimate did not account for possible participant attrition. Study inclusion criteria were 18–30 years old and fluent in English. There were no exclusionary criteria. Study procedures were approved by the Trinity College Institutional Review Board (ID: 3172). All methods were carried out in accordance with relevant guidelines and regulations. This research was not pre-registered.

We recruited participants through Prime Panels [29], an online survey recruitment platform that aggregates opt-in market research panels to enable data collection based on demographic quotas. After providing their informed consent, participants completed a Qualtrics questionnaire assessing their social media usage, sleep, and mental health characteristics. Participants received financial compensation for completing the study that depended on the platform used to access the survey. Data collection occurred between 20 February 2024 and 22 February 2024.

### 2.2. Measures

#### 2.2.1. Social Media Use

The volume of social media use was assessed using a single item developed by Levenson et al. [7] that asked participants to estimate the number of total minutes they used social media platforms per day on average for personal, non-work-related use, with response options of 0–30 min, 31–60 min, 61–120 min, and 121+ min.

The frequency of social media use was assessed using a questionnaire adapted from Levenson et al. [7] that asked participants to indicate their number of weekly visits to 11 types of social media platforms: Facebook, YouTube, X (formerly known as Twitter), Instagram, Snapchat, Reddit, Pinterest, LinkedIn, TikTok, Yik Yak, and messenger apps (including Facebook Messenger, WhatsApp, WeChat, text messaging, etc.). Each platform was rated on a seven-point scale with response options: I do not use this platform, less than once a week, 1–2 days a week, 3–6 days a week, about once per day, 2–4 times per day, and 5+ times per day.

Emotional investment in social media was assessed using an adaptation of the 10-item Social Media Use Integration Scale [30]. Consistent with earlier studies [11], we changed the phrasing of these questions to focus on overall emotional investment in social media (e.g., “Using social media is part of my everyday routine.”). The items were rated on a six-point scale from 1 (strongly disagree) to 6 (strongly agree) and summed/averaged to create a single composite variable representing overall emotional investment in social media (α = 0.868).

#### 2.2.2. Sleep Characteristics

Poor sleep quality was assessed using two questionnaires. The 10-item Pittsburgh Sleep Quality Index (PSQI) [31] asked participants to summarize their sleep habits during the past month in terms of subjective quality, sleep latency, sleep duration, habitual sleep efficiency, sleep disturbances, daytime alertness, and use of sleep medication. Scores on each dimension are summed to create a global score that represents poor sleep quality (α = 0.632). The 7-item Insomnia Severity Index (ISI) [32] asked participants to rate the extent to which they have been experiencing insomnia symptoms during the past two weeks. Responses to these items were used to calculate an overall insomnia severity score that ranged from 0 to 28, with higher scores representing greater insomnia symptom severity (α = 0.858).

#### 2.2.3. Negative Social Comparison on Social Media

We used the 7-item Negative Social Media Comparison Scale [33] to assess the extent to which participants negatively compared themselves to others on social media (e.g., “When I use social media, I feel like other people’s lives are better than mine”). Items were rated on a six-point scale from 1 (strongly disagree) to 6 (strongly agree) and summed/averaged to create a single composite variable (α = 0.893).

#### 2.2.4. Pre-Sleep Arousal

We used the 16-item Pre-Sleep Arousal Scale [34] to the degree to which participants generally experience symptoms of cognitive and somatic arousal during bedtime while they are trying to fall asleep (e.g., “Review or ponder the events of the day” and “Shortness of breath or labored breathing”). Items were rated on a five-point scale from 1 (not at all) to 5 (extremely) and summed/averaged to create a single composite variable (α = 0.917).

#### 2.2.5. Demographic Characteristics

Participants completed a demographic questionnaire assessing their age (continuous), gender (categorical: female, male, and non-binary), race/ethnicity (categorical: White, Black or African American, Hispanic or Latino, Asian or Asian American, American Indian or Alaska Native, Native Hawaiian or Pacific Islander, Multiple Races, or Another Race/Ethnicity), and educational attainment (ordinal: less than high school, high school or equivalent, some college and no degree, associate’s degree, bachelor’s degree, and graduate degree).

### 2.3. Data Analysis

We conducted our analyses using SPSS software (Version 29.0, IBM, Armonk, NY, USA). We tested Aim 1 by conducting a one-way multivariate analysis of variance with poor sleep quality and insomnia symptom severity as the dependent variables, frequency of social media use, volume of social media use, and emotional investment in social media as the independent variables, and age, gender, race, and educational attainment as covariates. We controlled for these demographic characteristics because of their known or potential associations with social media use [35] and sleep [36].

We tested Aim 2 by using the PROCESS macro for SPSS (Version 4.2) to test the strength and significance of the hypothesized indirect effects of emotional investment (IV) on poor sleep quality (DV) mediated by negative comparisons on social media (M1) and pre-sleep arousal (M2). Specifically, we used Model 6 of the PROCESS macro to test the serial mediation model of social media use (emotional investment and frequency of use) → negative comparisons on social media → pre-sleep arousal → sleep quality variables when controlling for demographic characteristics using bias-corrected bootstrapping with 10,000 resamples [37].

## 3. Results

### 3.1. Descriptive Analyses

Our final sample consisted of 830 of 849 participants (97.8%) who completed the survey and responded to the data quality question by committing to providing thoughtful answers to the survey questions. The sample ranged in age from 18 to 30 years (M = 24.2, SD = 3.9). Most participants identified as female (63.1% female, 35.9% male, 1.0% non-binary) and White (54.0% White, 28.1% Black or African American, 15.8% Hispanic or Latino, 4.9% Asian or Asian American, 3.0% American Indian or Alaska Native, 1.9% multiple races, 0.6% another race, 0.2% Native Hawaiian or Other Pacific Islander). The highest level of education completed by most participants was a high school diploma or equivalent (41.7%), with fewer participants completing less than high school (6.3%), some college but no degree (24.9%), an associate’s degree (9.8%), a bachelor’s degree (14.3%), or a graduate degree (3.0%).

Most participants reported using social media platforms for over half an hour daily (9.0% for 0–30 min, 26.7% for 31–60 min, 33.5% for 61–120 min, and 30.7% for 121+ min). In addition, nearly all participants reported the use of electronic devices after lights out; the mean, median, and modal response to this question was between 21–60 min of continued electronic device use after switching off the lights for sleep (6.3% not at all, 8.7% for 0–10 min, 15.5% for 11–20 min, 25.3% for 21–60 min, 19.4% for 1–2 h, and 24.8% for 2+ h).

Table 1 summarizes the average frequency of use for specific social media platforms. The most frequently used platforms in this sample were YouTube (M = 4.9, SD = 1.9), Instagram (M = 4.6, SD = 2.2), and TikTok (M = 4.6, SD = 2.4). The least frequently used platforms in this sample were Yik Yak (M = 1.4, SD = 1.2), LinkedIn (M = 2.0, SD = 1.6), and Reddit (M = 2.5, SD = 1.8).

Most participants reported some degree of insomnia symptoms, with 28.6% meeting the criteria for the absence of insomnia, 41.3% for sub-threshold insomnia, 24.2% for moderate insomnia, and 5.9% for severe insomnia. Most participants also exceeded the PSQI threshold score for poor sleep quality (75.4%). There was a moderate-strong correlation between continuous scores on the ISI and PSQI, *r* = 0.68, *p* < 0.001.

### 3.2. Aim 1: Association of Social Media Use and Sleep Characteristics

We tested whether the frequency, volume, and emotional investment in social media use were associated with sleep habits when controlling for age, gender, race/ethnicity, and educational attainment. Consistent with Hypothesis 1, we observed statistically significant effects of emotional investment in social media, F(2,820) = 41.62, *p* < 0.001, Wilks’ Λ = 0.908, partial η^2^ = 0.092, and frequency of social media use, F(2,820) = 12.55, *p* < 0.001, Wilks’ Λ = 0.970, partial η^2^ = 0.030, but not volume of social media use on the combined sleep-dependent variable, F(2,820) = 0.13, *p* = 0.43, Wilks’ Λ = 0.998, partial η^2^ = 0.002.

Parameter estimates indicated that this was attributable to associations of more frequent social media use, b = 0.68, SE = 0.14, 95CI = 0.41, 0.95, *p* < 0.001, and greater emotional investment in social media, b = 0.94, SE = 0.13, 95CI = 0.68, 1.21, *p* < 0.001, with poorer sleep quality, and to associations of more frequent social media use, b = 0.56, SE = 0.21, 95CI = 0.14, 0.98, *p* = 0.009, and greater emotional investment in social media with greater insomnia severity, b = 1.87, SE = 0.21, 95CI = 1.46, 2.28, *p* < 0.001.

### 3.3. Aim 2: Mechanisms Linking Social Media Use and Sleep Characteristics

We used serial mediation analysis to test whether the association between social media use (emotional investment and frequency of use) and poor sleep quality was mediated by negative comparisons of social media and pre-sleep arousal. These models simultaneously tested the simple mediating effect of negative comparisons on social media, the simple mediating effect of pre-sleep arousal, and the serial mediating effect of negative comparisons on social media and pre-sleep arousal.

#### 3.3.1. Sleep Quality

Figure 2 shows the model results for emotional investment in social media and poor sleep quality. Consistent with Hypothesis 2a, we observed that the association between emotional investment in social media and sleep quality was mediated by negative comparisons on social media, β = 0.06, SE = 0.02, 95CI = 0.03, 0.09, and pre-sleep arousal, β = 0.17, SE = 0.02, 95CI = 0.13, 0.21. Consistent with Hypothesis 2b, we also observed evidence for a serial mediation effect of emotional investment in social media → negative comparisons on social media → pre-sleep arousal → sleep quality, β = 0.04, SE = 0.01, 95CI = 0.03, 0.06.

Figure 3 shows the model results for the frequency of social media use and poor sleep quality. Also consistent with Hypothesis 2a, we observed that the association between the frequency of social media use and sleep quality was mediated by negative comparisons on social media, β = 0.03, SE = 0.01, 95CI = 0.01, 0.05, and pre-sleep arousal, β = 0.07, SE = 0.02, 95CI = 0.05, 0.10. Consistent with Hypothesis 2b, we also observed evidence for a serial mediation effect of frequency of social media use → negative comparisons on social media → pre-sleep arousal → sleep quality, β = 0.03, SE = 0.01, 95CI = 0.02, 0.05.

#### 3.3.2. Insomnia Severity

Figure 4 shows the model results for emotional investment in social media and insomnia severity. Consistent with Hypothesis 2a, we observed that the association between emotional investment in social media and insomnia severity was mediated by negative comparisons on social media, β = 0.08, 95CI = 0.05, 0.11, and pre-sleep arousal, β = 0.21, 95CI = 0.16, 0.26. Consistent with Hypothesis 2b, we also observed evidence for a serial mediation effect of emotional investment in social media → negative comparisons on social media → pre-sleep arousal → insomnia severity, β = 0.05, 95CI = 0.04, 0.07. When accounting for these indirect effects, the direct effect of emotional investment in social media on insomnia severity was no longer significant, β = 0.02, 95CI = −0.05, 0.08.

Figure 5 shows the model results for frequency of social media use and insomnia severity. Also consistent with Hypothesis 2a, we observed that the association between the frequency of social media use and insomnia severity was mediated by negative comparisons on social media, β = 0.04, SE = 0.01, 95CI = 0.02, 0.06, and pre-sleep cognitive arousal, β = 0.09, SE = 0.02, 95CI = 0.06, 0.13. Consistent with Hypothesis 2b, we also observed evidence for a serial mediation effect of frequency of social media use → negative comparisons on social media → pre-sleep arousal → insomnia severity, β = 0.04, SE = 0.01, 95CI = 0.03, 0.06.

## 4. Discussion

The current study examined the relationship between social media use and sleep in a sample of emerging adults (ages 18–30) in the United States. We observed that more frequent social media use and greater emotional investment in social media were associated with poorer sleep quality and greater insomnia severity; the volume of social media use was not associated with sleep outcomes. This study also aimed to investigate negative social comparison and pre-sleep cognitive arousal as potential mediators of the association between social media use and sleep. We observed that the association between emotional investment in social media and poor sleep outcomes was mediated by negative social comparison and pre-sleep cognitive arousal. Finally, we observed evidence for a serial indirect effect of emotional investment in social media on poor sleep via negative social comparison and pre-sleep cognitive arousal.

The first aim of this study was to test the association between social media use and poor sleep in a sample of emerging adults. Consistent with our hypothesis and earlier studies [10,11], we found that more frequent social media use and greater emotional investment in social media were associated with poorer sleep quality and greater insomnia severity. These observations contribute to a significant body of research linking social media use with poor sleep [4,6]. Future studies are needed to identify the specific characteristics of sleep and circadian rhythms adversely impacted by social media use. For example, the observed associations between social media use and poor sleep may reflect an adverse impact of social media use on sleep duration, sleep efficiency, and/or the timing of sleep-wake behaviors.

We also observed that emotional investment in social media was a stronger predictor of poor sleep than other measures of social media use. Although more frequent social media use and greater emotional investment in social media were both associated with poorer sleep, the effect size statistic was more than three times larger for emotional investment in social media than for frequency of social media use. This is consistent with the possibility that emotional investment is a key dimension of social media use that more directly relates to its negative impact on sleep than other measures, such as the frequency or volume of use [11]. Greater emotional investment in social media reflects the extent to which an individual is attached to social media, prefers using social media to connect and communicate with others, and has integrated social media use into their daily life [30]. In theory, this greater emotional investment may lead individuals to experience difficulty disengaging physically and cognitively from social media at bedtime, subsequently disrupting their nightly sleep. If these results were replicated in subsequent studies, then it would suggest that emotional investment in social media may represent a more effective target for developing interventions designed to improve sleep in young adults.

The second aim of this study was to test negative social comparison and pre-sleep cognitive arousal as potential mechanisms linking social media use and sleep. Consistent with our hypothesized model, we found that negative social comparison and pre-sleep cognitive arousal mediated the associations of more frequent social media use and greater emotional investment in social media with sleep outcomes. Specifically, we observed a simple mediation effect for negative social comparison, a simple mediation effect for pre-sleep cognitive arousal, and a serial mediation effect for negative social comparison and pre-sleep cognitive arousal.

Greater emotional investment in social media and more frequent social media use were associated with a tendency for young adults to engage in negative social comparisons to others when using social media. In turn, engaging in negative social comparison when using social media was associated with increased pre-sleep cognitive arousal, which subsequently resulted in worse sleep quality and greater insomnia severity. In theory, negatively comparing one’s overall life, social experiences, or physical appearance to the idealized and selective self-presentations of other social media users may lead to a state of heightened emotional, cognitive, and physiological arousal that prevents young adults from falling or staying asleep. This possibility is consistent with earlier research linking the tendency to engage in negative social comparison when using social media with poorer self-esteem and well-being in young adults [11,19,38]. Future studies are needed to determine which specific aspects of negative social comparison are most strongly predictive of pre-sleep cognitive arousal and poor sleep outcomes. Future experimental studies are also justified to test whether engaging in negative social comparison using social media impacts physiological measures of pre-sleep arousal, such as heart rate or blood pressure.

Notably, we observed a direct mediating effect of pre-sleep arousal unrelated to negative social comparison. Although we hypothesized that negative social comparison was an important process linking greater social media use with increased pre-sleep cognitive arousal, this observation suggests that additional social psychological mechanisms are at play. Other candidate mechanisms linking social media use with pre-sleep cognitive arousal that should be explored in future studies include exposure to blue light from electronic devices [39] and experiencing distress, anxiety, or fear of missing out when disengaging from social media at bedtime [9].

### Strengths, Limitations, and Future Directions

The current study’s strengths include its relatively large sample of more than 800 young adults, its use of previously validated self-report measures of sleep characteristics, pre-sleep arousal, negative social comparison, and emotional investment in social media, and its statistical testing of a theoretical model linking social media use with poor sleep. However, several limitations of this study provide an opportunity for future research.

First, this study was cross-sectional and, therefore, unable to address the possibility of reverse causation [40]. Specifically, the observed association of sleep and social media use may be attributable to poor sleepers choosing to engage in more social media use at night as a potential sleep aide or a distraction from being unable to fall asleep [41]. This possibility is consistent with the theory of compensatory internet use, which proposes that social media use is driven by the desire to reduce negative effects and fulfill unmet social and psychological needs [42]. Similarly, poor sleepers may engage in more social media use because a lack of sleep depletes the self-regulatory resources needed to disengage from social media use at nighttime. This possibility is consistent with the socio-cognitive model of internet addiction’s assertion that harmful social media use habits arise because individuals with poorer self-regulation are more susceptible to the psychologically rewarding nature of social media use [43]. This potential explanation is also supported by the observed association of poor sleep and greater emotional investment in social media among young adults in the current study. Indeed, longitudinal studies of social media use and depression symptoms have generally found stronger evidence for social media use as a consequence, rather than an antecedent, of depression [40]. Future studies could address the issue of causation by using intensive longitudinal designs that would provide evidence for the temporality of the association between social media use and sleep or by using experimental methods that would allow for direct manipulation of sleep and/or social media use.

Second, this study assessed sleep and social media habits using self-report questionnaires that required participants to self-report their typical behaviors and experiences on average during the past month or several weeks. Our self-report measures of sleep and social media habits may have been imprecise, influenced by social desirability, susceptible to cognitive biases, or influenced by third-factor variables like neuroticism. Future research could address these limitations by using actigraphy or polysomnography to measure sleep and using more objective methods to assess social media use, such as screen time data.

Third, this study was focused on young adults because of the well-documented sleep deficits this population experiences [2,3]. However, there is also evidence linking social media use with poor sleep across the lifespan. Indeed, Perez et al. [44] found that the strength of the association between social media use and sleep increased with age. Thus, it remains unclear whether negative social comparison and/or pre-sleep arousal are also important mechanisms linking social media use and sleep in older populations. Future studies should continue to investigate the mechanisms linking social media use and poor sleep across the lifespan.

Fourth, this study did not address possible third-factor explanations, such as neuroticism, depression, loneliness, and perceived stress, that may impact both social media use and sleep. These factors should be assessed and controlled for in subsequent research.

Finally, it is important to acknowledge that the observed effect sizes of the mediation pathways were relatively small. Thus, whether these findings have additional practical significance beyond their theoretical relevance is unknown. Future research could address this limitation by testing whether directly manipulating these hypothesized mediators impacts sleep.

## 5. Conclusions

This study contributes to a significant body of evidence linking social media use with poor sleep during young adulthood. We replicated earlier studies by demonstrating that greater emotional investment may represent a key dimension of social media use that worsens sleep in young adults. We also extended earlier findings by providing cross-sectional evidence for negative social comparison and pre-sleep cognitive arousal as two potential mechanisms linking social media use and poor sleep outcomes. Overall, these findings emphasize that excessive social media use may interfere with the development of healthy sleep patterns during young adulthood. Given the importance of good sleep for health and well-being across the lifespan, future research is needed to explore whether interventions designed to reduce young adults’ emotional investment in social media could potentially improve sleep outcomes.

## Figures and Tables

**Figure 1 behavsci-14-00794-f001:**
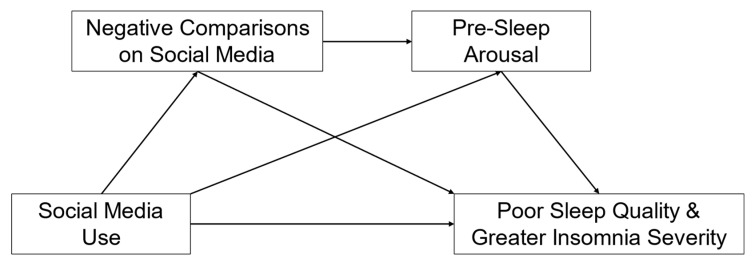
Hypothesized serial mediation model of the mechanisms linking social media use and sleep.

**Figure 2 behavsci-14-00794-f002:**
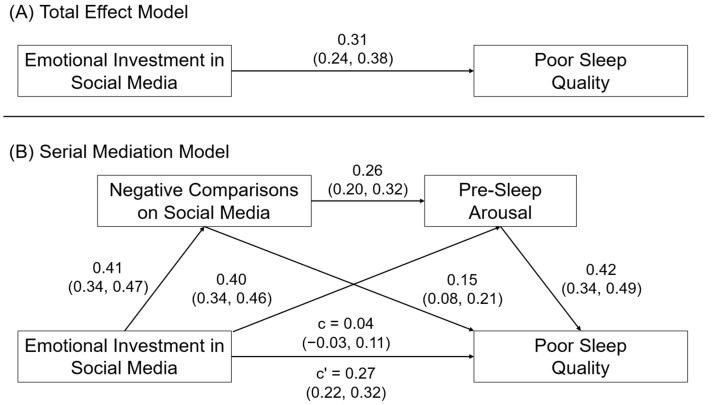
(**A**) Direct and (**B**) serial mediation effects of emotional investment in social media on sleep quality. Relationships are shown as completely standardized effects and 95% confidence intervals.

**Figure 3 behavsci-14-00794-f003:**
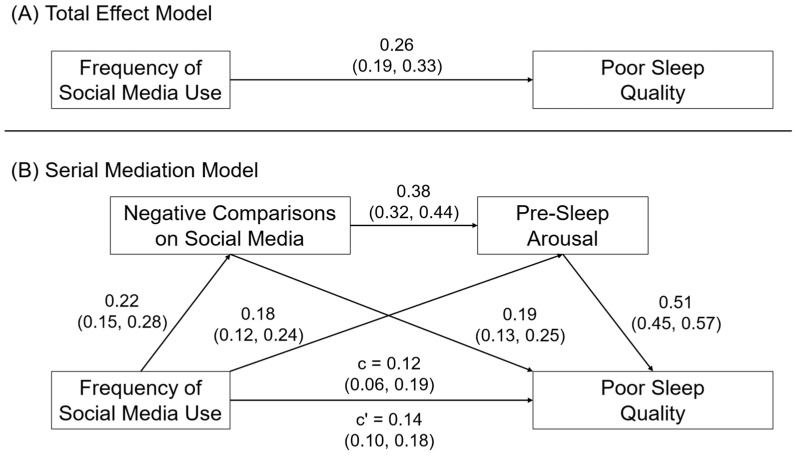
(**A**) Direct and (**B**) serial mediation effects of frequency of social media use on sleep quality. Relationships are shown as completely standardized effects and 95% confidence intervals.

**Figure 4 behavsci-14-00794-f004:**
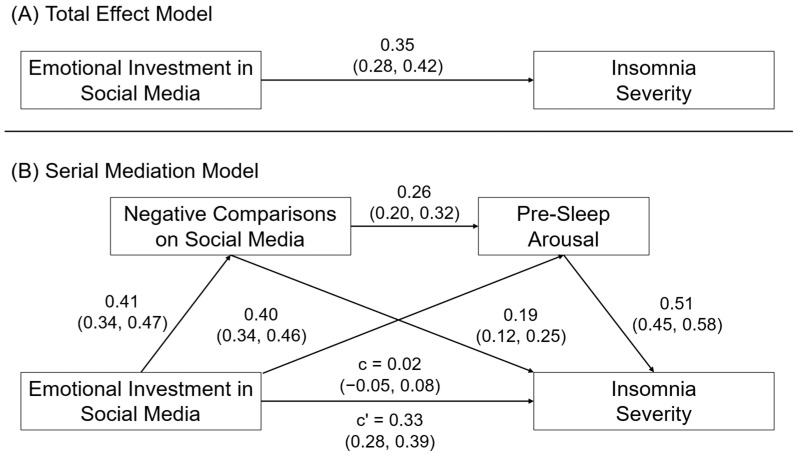
(**A**) Direct and (**B**) serial mediation effects of emotional investment in social media on insomnia severity. Relationships are shown as completely standardized effects and 95% confidence intervals.

**Figure 5 behavsci-14-00794-f005:**
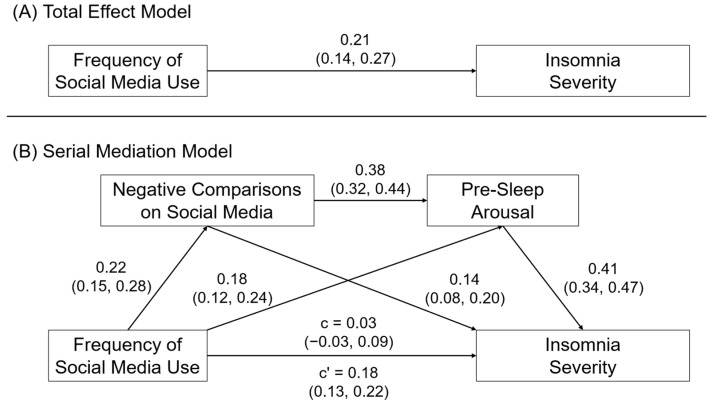
(**A**) Direct and (**B**) serial mediation effects of frequency of social media use on insomnia severity. Relationships are shown as completely standardized effects and 95% confidence intervals.

**Table 1 behavsci-14-00794-t001:** Demographic and sleep characteristics of the sample (N = 830).

Characteristic	Mean (SD) or % (n)
Age (years)	24.2 (3.9)
Gender	
Female	63.1 (524)
Male	35.9 (298)
Non-Binary	1.0 (8)
Race and Ethnicity	
White	54.0 (448)
Black or African American	28.1 (233)
Hispanic or Latino	15.8 (131)
Asian or Asian American	4.9 (41)
American Indian or Alaska Native	3.0 (25)
Native Hawaiian or Pacific Islander	0.2 (2)
Another Race or Ethnicity	0.6 (5)
Multiple Races	1.9 (16)
Educational attainment	
Less than high school	6.3 (52)
High school or equivalent	41.7 (346)
Some colleges, no degree	24.9 (207)
Associate’s degree	9.8 (81)
Bachelor’s degree	14.3 (119)
Graduate degree	3.0 (25)
Sleep characteristics	
Pittsburgh Sleep Quality Index	8.6 (3.9)
Insomnia Severity Scale	11.3 (6.1)

Note. SD = Standard Deviation.

## Data Availability

The data for this study are available at: https://osf.io/ed7kr/ (accessed on 14 June 2024).

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
