# Peer review of "Mechanisms Linking Social Media Use and Sleep in Emerging Adults in the United States"

_behavsci, 2024, doi:10.3390/bs14090794_

Round 1

Reviewer 1 Report

Comments and Suggestions for Authors

Thank you for the opportunity to review this article. Even though it does not examine anything entirely new, the article is clearly formulated, comprehensible and without any obvious shortcomings. Above all, the theory- and hypothesis-driven approach is very positive.

I have listed a few small points below that could help to improve the truly good paper.

·         The link to social media use should be present in the keywords.

·         I would recommend adding references for the statement “social media use is one common cause of poor sleep”.

·         Introduction: It would be a little more precise to specify social media use (where differences are known). For example, whether the contexts involve active or passive consumption and which consumption (social networks or other).

·         A short explanation about the symptoms of insomnia and fatigue as well as for the construct  fear of missing out would be helpful.

·         I would recommend adding a reference for TikTok.

·         Some questions that could supplement section 2.1: Was a drop-out rate taken into account in the power planning? Is there no number/file number for the ethics vote? Has the study been preregistered?

·         The abbreviations PSQI and ISI were not introduced.

·         Were there no non-binary people in the sample?

·         This is a repeat and can be deleted: “We tested Aim 2 by using the PROCESS macro for SPSS (Version 4.2)” and “we used Model 6 of the PROCESS macro”.

·         The specific numbering of the hypotheses only takes place in the results section. For a better understanding, I recommend doing this in the corresponding previous section.

Author Response

Comment 1: The link to social media use should be present in the keywords.

Response: We added social media use to the keywords.

Comment 2: I would recommend adding references for the statement “social media use is one common cause of poor sleep”.

Response: We added a supporting reference for this statement on Page 1.

Comment 3: Introduction: It would be a little more precise to specify social media use (where differences are known). For example, whether the contexts involve active or passive consumption and which consumption (social networks or other).

Response: We revised the Introduction section to be specific about the type of social media use assessed in earlier studies when possible.

Comment 4: A short explanation about the symptoms of insomnia and fatigue as well as for the construct fear of missing out would be helpful.

Response: We added these explanations on Page 2.

Comment 5:  I would recommend adding a reference for TikTok.

Response: We added a supporting reference for TikTok on Page 2.

Comment 6: Some questions that could supplement section 2.1: Was a drop-out rate taken into account in the power planning? Is there no number/file number for the ethics vote? Has the study been preregistered?

Response: We added this information to Section 2.1.

Comment 7: The abbreviations PSQI and ISI were not introduced.

Response: We revised the manuscript to introduce these abbreviations on Page 4.

Comment 8: Were there no non-binary people in the sample?

Response: We revised the text and table to clarify that our sample included 1.0% non-binary participants.

Comment 9: This is a repeat and can be deleted: “We tested Aim 2 by using the PROCESS macro for SPSS (Version 4.2)” and “we used Model 6 of the PROCESS macro”.

Response: We deleted the repeat text.

Comment 10: The specific numbering of the hypotheses only takes place in the results section. For a better understanding, I recommend doing this in the corresponding previous section.

Response: We added specific numbering of our hypotheses in the Aims and Hypotheses subsection of the Introduction.

Reviewer 2 Report

Comments and Suggestions for Authors

This is an interesting study investigating the relationship between social media use and sleep outcomes among young adults aged 18-30. The authors hypothesize that emotional investment in social media and the frequency of social media use are associated with poorer sleep quality and greater insomnia severity. The study further explores negative social comparison and pre-sleep cognitive arousal as mediators of this relationship. Overall, the paper is well-written and the study addresses a relevant and timely issue. I agree that it may contribute well to the literature. I have several comments to improve the manuscript further:

1. First, the study primarily focuses on negative social comparison and pre-sleep cognitive arousal as mediators. However, other mechanisms, such as exposure to blue light from screens and fear of missing out (FOMO). Perhaps, these mechanisms should be acknowledged in the discussion.

2. The hypotheses could be more clearly stated in the introduction

3. The effect sizes reported for some of the mediation pathways are relatively small. The authors should provide a more detailed discussion of the practical significance of these findings

Ferguson, C. J. (2016). An effect size primer: A guide for clinicians and researchers. In A. E. Kazdin (Ed.), Methodological issues and strategies in clinical research (4th ed., pp. 301–310). American Psychological Association. https://doi.org/10.1037/14805-020

4. The results could be enhanced by providing confidence intervals for all effect sizes

5. I really appreciate the authors careful acknowledgement of the potential reverse causation. I believe that this issue, specifically the idea that social media can serve as a coping strategy, deserves its own dedicated paragraph in the discussion section. This is important to be elaborated further especially given that this is a cross-sectional study with a mediation model. A proper discussion, rather than a brief acknowledgment, would strengthen the overall argument and provide a more nuanced view of the findings. For instance, please see the following relevant paper that discuss about reverse causation perspective

Does social media use increase depressive symptoms? A reverse causation perspective. Frontiers in Psychiatry, 12, 641934.

6. The study should justify the choice of control variables and address the omission of key psychological factors, such as neuroticism, which could significantly influence the relationship between social media use and sleep outcomes.

Statistical control requires causal justification. Advances in Methods and Practices in Psychological Science, 5(2), 25152459221095823.

Author Response

Comment 1: First, the study primarily focuses on negative social comparison and pre-sleep cognitive arousal as mediators. However, other mechanisms, such as exposure to blue light from screens and fear of missing out (FOMO). Perhaps, these mechanisms should be acknowledged in the discussion.

Response: We added several sentences to the discussion section on Page 10 to acknowledge these other mechanisms and the need to examine them in subsequent studies.

Comment 2: The hypotheses could be more clearly stated in the introduction

Response: We revised our hypotheses for clarity on Page 3.

Comment 3: The effect sizes reported for some of the mediation pathways are relatively small. The authors should provide a more detailed discussion of the practical significance of these findings.

Response: We added several sentences to the discussion section on Page 11 acknowledging the relatively small effect sizes of some mediation pathways.

Comment 4: The results could be enhanced by providing confidence intervals for all effect sizes.

Response: We added confidence intervals to the main text on Page 6 and to each of the figures.

Comment 5: I really appreciate the authors careful acknowledgement of the potential reverse causation. I believe that this issue, specifically the idea that social media can serve as a coping strategy, deserves its own dedicated paragraph in the discussion section. This is important to be elaborated further especially given that this is a cross-sectional study with a mediation model. A proper discussion, rather than a brief acknowledgment, would strengthen the overall argument and provide a more nuanced view of the findings. For instance, please see the following relevant paper that discuss about reverse causation perspective: Does social media use increase depressive symptoms? A reverse causation perspective. Frontiers in Psychiatry, 12, 641934.

Response: Thank you for this suggestion. We added a more detailed discussion of the possibility of reverse causation on Page 11.

Comment 6: The study should justify the choice of control variables and address the omission of key psychological factors, such as neuroticism, which could significantly influence the relationship between social media use and sleep outcomes.

Response: We added sentences to justify our use of control variables on Page 5. We also added several sentences to the discussion section on Page 11 to acknowledge the role of potential third factors, such as neuroticism, that could influence the relationship between social media use and sleep.

Round 2

Reviewer 2 Report

Comments and Suggestions for Authors

The authors have addressed all my comments well. The manuscript is ready for publication